# Misperception of the facial appearance that the opposite-sex desires

**David I. Perrett** \*, **Iris J. Holzleitner**  ¤a, **Xue Lei** ¤b

School of Psychology & Neuroscience, University of St Andrews, St Andrews, Fife, Scotland, United Kingdom

¤a Current address: School of Social Sciences, University of the West of England, Bristol, United Kingdom
¤b Current address: School of Business Administration, Zhejiang University of Finance and Economics, Hangzhou, China

* dp@st-andrews.ac.uk

**Data Availability Statement:** The data that support the findings of this study are openly available on the Open Science Framework at https://osf.io/3rkxz/?view_only=

## Abstract

Thin and muscular have been characterised as body shape ideals for women and men, respectively, yet each sex misperceives what the other sex desires; women exaggerate the thinness that men like and men exaggerate the muscularity that women like. Body shape ideals align with stereotypic perceptions of femininity in women and masculinity in men. The present study investigates whether misperception of opposite-sex desires extends to femininity/masculinity in facial morphology. We used interactive 3D head models to represent faces varying in sexual dimorphism. White European heterosexual men and women were asked to choose their own and ideal face shape, the ideal shape of a short- and a long-term partner, and the face shape they thought the opposite sex would most like for a short- and a long-term partner. Women overestimated the facial femininity that men prefer in a partner and men overestimated the facial masculinity that women prefer in a partner. The discrepancy between own and ideal sexual dimorphism (an index of appearance dissatisfaction) covaried with by the misperception of what the opposite-sex desires. These results indicate misperception of opposite-sex facial preferences and that mistaken perceptions may contribute to dissatisfaction with own appearance.

## Introduction

Research on body image reveals a misperception of the body shape desired by others which may contribute to dissatisfaction with own appearance. This study assesses whether there is a similar misperception in the sexual dimorphism of facial shape desired by others and whether the degree of overestimation predicts dissatisfaction with own facial appearance. The study hypotheses were not pre-registered.

### Thin and muscular body ideals

In Western countries, the ideal female figure is thin (e.g., [1]) while the ideal male figure is lean and muscular (e.g., [2]). The drive to attain an ideal body shape leads to unhealthy behaviour including excessive dieting to lose weight in women [3] and use of anabolic steroids to

46a7dd9adae048218c328ab9a912b331 The data are also available at https://figshare.com/articles/dataset/Misperception_of_the_desirability_of_sexual_dimorphism/26424307?file=48071893.

**Funding:** The author(s) received no specific funding for this work.

**Competing interests:** The authors have declared that no competing interests exist.

develop muscles in men [4,5]. While there are pressures from the media, peers and family that contribute to individuals' dissatisfaction with their appearance [6–8], a further contributing factor to body dissatisfaction may be the misperception of opposite-sex preferences.

Women overestimate the thinness that men desire in a partner [9–12] and men overestimate the heaviness and muscularity that women desire in a partner [9,11–14]. The nature and extent of these misperceptions are important as they may contribute to the adoption of unhealthy attitudes and behaviour. For example, in women, the extent of misperception of male desires predicts the extent of unhealthy eating attitudes and disordered eating behaviour [10].

## Body shape desires reflect stereotypic concepts of femininity and masculinity

Thinness in women and muscularity in men are attributes that are stereotypically linked to concepts of femininity and masculinity. Guy et al. [15] found that, for USA college students, a thin woman's body shape was sex-typed feminine whereas a lean muscular male body was sex-typed masculine. Indeed, from a very early age, body thinness is associated with femininity. Pre-school girls aged 3–5 years are emotionally invested in the thin is positive stereotype and associate female thinness with feminine traits [16]. Indeed, for both girls and boys, stereotypic feminine traits (e.g., likes children) were more strongly associated with a thinner female figure than with a fatter one [17]. Conversely, muscular physiques are viewed by men as being masculine (e.g., [18]). For example, McCreary et al. [19] concluded that men believe "those with a greater degree of muscularity are more masculine and that gaining muscularity increases one's masculinity."

The misperception of opposite-sex desires extends to other features of sexual dimorphism. Women overestimate the bust-to-waist ratio that men find attractive [20]. Perhaps consequently most women would prefer to have bigger breasts [21]. Since misperception is widespread and frequently relates to feminine and masculine attributes, there is reason to anticipate that there will be a misperception of the sexual dimorphism in face shape that the opposite-sex desires.

## Sexual dimorphism in face shape

Masculinity in male face shape has variable effects on attractiveness [22] with some studies reporting that women prefer masculinised male faces while other studies find no effect or a preference for feminised male faces.

In industrial populations, male facial masculinity relates strongly to a dominant appearance [23] which is an important attribute for intra-sexual competition amongst men. One indication that male masculinity might be a sign of good quality, at least in industrialised societies [24], comes from consideration of competition in the 'mating market' [25]. Good quality from this limited perspective refers to individuals that are desired as a partner, and who can outcompete rivals in their own quest to secure a desirable partner. Multiple attributes contribute to mate quality (including resources, youth and physical attractiveness). Several studies have reported that women who consider themselves as more attractive than average prefer a higher degree of male facial masculinity than women that consider themselves less attractive [26–31] although Alharbi et al. [32] reported a null finding in an Arab sample). Hence, facial masculinity in men can be considered as a sign of quality in the mating market.

Such individual differences in masculinity preferences can arise if both facial masculinity in men and attractiveness in women are considered signs of good quality. Competition for partners predicts that a high-quality individual will be able to form a stable partnership with

another individual of similar high quality. Reciprocally individuals of lower quality will tend to form partnerships with similar quality individuals. Hence, competition results in assortative mating for quality where like pairs with like. Such an argument considers only one dimension of quality; real pairings will be multifactorial and will involve trade-offs between attributes. Furthermore, what constitutes a mark of quality varies between populations [33].

There is less dispute that sexually dimorphic characteristics in women such as femininity of body shape [34] and face shape [35] and even hair length [36] increase attractiveness.

As argued above, an indication that feminine characteristics in women are a sign of quality (at least in Western societies, [33,37]) comes from consideration of competition. In a similar manner to that described above, men who rate themselves as attractive prefer a higher degree of female facial femininity than men who consider themselves as less attractive [27,31,38,39]. Hence, from the perspective of competition, sexual dimorphism in face shape can be considered a desirable quality for both men and women in industrial societies.

Independent of how facial dimorphism might signal quality, it is evident that dimorphism influences attractiveness in both men and women. Hence, people may be concerned about the degree to which their facial masculinity or femininity conforms to ideals. Individuals may aspire to higher levels of dimorphism than they themselves exhibit, since this would make them more competitive in acquiring a high-quality mate. As noted for sexually dimorphic aspects of body shape, the opposite sex is often predicted to desire a more exaggerated appearance than they prefer. In the same way, the dimorphism in facial appearance that is predicted to be desired by the opposite sex may be exaggerated. Women may have an exaggerated notion of the facial femininity that men desire, and men may have an exaggerated notion of the facial masculinity that women desire.

## Dissatisfaction with appearance

In research on body image, participants are often presented with a range of body shapes along a continuum from underweight (emaciated) to overweight (obese) and asked to choose a shape which best fits their current body shape, and their ideal body shape. A measure of perceived dissatisfaction with own physiognomy is derived from difference between perception of ideal body shape and own body shape [40–42]. This discrepancy (own minus ideal) is associated with a person's wellbeing and self-esteem: Kostanski and Gullone [43] reported that participants with a high level of dissatisfaction with their perceived body image had low self-esteem and high levels of anxiety and depression.

One can derive a similar index of dissatisfaction for facial dimorphism. This index can be computed as the difference between the participant's ideal facial dimorphism and the participant's own facial dimorphism (ideal minus own dimorphism). This index will be used here as a proxy for dissatisfaction with facial appearance. In an analogous way to body image dissatisfaction, we predict that many participants will exhibit dissatisfaction with own facial appearance being lower in sexual dimorphism than the participant's ideal.

Misperception of the desires of others may be a contributing factor to dissatisfaction with own appearance. Lei and Perrett [12] found that the extent of body image dissatisfaction in both men and women was predicted by the misperception of what the opposite sex desired in body shape. Here we investigate whether misperception of facial dimorphism desired by the opposite sex predicts dissatisfaction with facial dimorphism.

Most men choose an ideal body shape that is heavier and more muscular than how they perceive their own body to be. Reciprocally, women choose an ideal that is skinnier than what they perceive their own body to be (e.g., [12]). If women have a high body mass, then they are likely to have a greater discrepancy with their ideal body mass index. Hence, in investigating

dissatisfaction in facial morphology it is important to control for the participant's own facial morphology.

## Relationship context

A factor affecting mate choice preferences is relationship context. Long-term relationships come with direct benefits (to the participant) of parental investment, resources and safety, whereas short-term relationships can have indirect genetic benefits (to the participant's off-spring) but only temporary resources [44]. Different partner characteristics are therefore desired more in different contexts [45].

Preference for sexual dimorphism seems to be stronger in contexts where investment in a relationship is deemed less important, making relationship type a crucial distinction when investigating mate preferences. For both men and women, facial attractiveness has been shown to depend on the relationship context considered, with greater levels of sexual dimor-phism preferred for short-term relationships compared to long-term relationships [39,46,47]. Moreover, greater exaggeration in body-shape ideals has been found in judgements made for short-term relative to long-term relationship contexts [12]. Therefore, it is anticipated that exaggeration of sexual dimorphism in face shape desired by another will be greatest in short-term relationship contexts.

## Hypotheses

The general prediction is that men and women will overestimate the level of sexual dimor-phism in facial appearance that is desired by the opposite sex. To test this, we measure the male face shape that participant women desire and compare this to the face shape that men predict women will desire. Reciprocally, we measure the female face shape that participant men desire and compare this to the face shape that women predict men will desire.

The following hypotheses will be tested:

(H1) The level of facial masculinity that men predict women desire will be higher than the masculinity women actually prefer.

(H2) Men's misperception will be greater in a short- than a long-term relationship context.

(H3) The level of facial femininity that women predict men desire will be higher than the femi-ninity men actually prefer.

(H4) Women's misperception will be greater in a short- than a long-term relationship context.

(H5) The facial masculinity men think is ideal will be higher than the masculinity they think corresponds to their own face shape.

(H6) The facial femininity women think is ideal will be higher than the femininity they think corresponds to their own face shape.

(H7) The size of misperception of opposite-sex desires will covary with men's and women's dissatisfaction with their own face shape (defined as the difference between ideal and own facial dimorphism).

## Materials and methods

The work was approved by the University Teaching and Research Ethics Committee Ethics Committee of the University of St Andrews (PS13176). All participants gave informed consent

prior to commencing the experiment. Participants were given a Participant Information Sheet and were required to fill in an on-line consent form before proceeding to the experiment proper.

## Nomenclature

To disambiguate participants and stimuli, the participants will be referred to as *men* and *women*, and the stimuli will be referred to as *male* and *female*. Sexual dimorphism preferences will be named *predictions* when participants predict what the opposite sex prefer, and they will be named *preferences* for what participants judge most attractive in the opposite sex. Sexual dimorphism is referred to as *masculinity* when referring to male stimuli and *femininity* when referring to female stimuli.

## Participants

**Sample size considerations.** Lei and Perrett [12] reported misperception of body shape (body mass index) preference between men and women with effect sizes ranging between d = 0.4 and d = 0.9. If we assume that misperception of facial shape preferences has a similar range in effect size we can estimate an appropriate sample size. With a power of 80%, an alpha error of 0.05 and an effect size of d = 0.4, G-power suggested a sample size of 76 men and 76 women.

A sample of men and women (sex as recorded on legal/official documents) were recruited via Prolific. Recruitment started on 17th January 2021 at 15:18 and ended 18:35 of the same day. Pre-screening criteria were applied, which included age from 18 to 26 years old, hetero-sexual orientation, UK residency and White ethnicity. All participants received £1.50 as a reward for their participation (Median duration 9.3 minutes), a rate of £9.6 per hour that was considered good by Prolific.com at the time of experiment. One hundred and fifty-three participants were recruited including 77 men and 76 women. Exclusions were made based on participants not completing the task (n = 1), reporting sexual orientation <4 on a 1–7 sexual orientation questionnaire (n = 4) or choosing a "wish not to answer" option (n = 4). This left a sample of 72 men (age: M = 21.72, SD = 2.67, range 18–26) and 72 women (age: M = 22.28, SD = 2.69, range 18–27).

**Stimuli.** The study used 3D stimuli. The stimuli were two male and two female 'base' faces. Each of the male base faces was a composite or average of the shape, colour and texture of four male faces. Likewise, each of the female base faces was a composite of shape, colour and texture of four female faces. Composite stimuli were employed because they are more representative (of the category of faces from which they were constructed) than the original faces [48]. Additionally, the composite images cannot be recognised as known individuals.

These female and male composites or base faces were then feminised and masculinised in 3D shape based on the difference between an average of 50 male faces and an average of 68 female faces [49]. Fig 1 upper shows the shape transformation of the average male face rather than a base face. Likewise Fig 1 lower shows the transformation of the average female face. Transformation was achieved by applying or subtracting the linear difference between the average male face shape and female face shape to target face (i.e. average faces for illustration and each individual composite or base face for stimuli. The distance from the average female face shape to the average male face shape was equivalent to a +100% increase in masculinity. Thus, each of the 118 faces contributed to the definition of the dimorphism vector along which base faces were manipulated. Importantly, we compared the sexual dimorphism from a second independent set of 40 male and 40 female faces [50]. The sexual dimorphism vectors from the within-set and the independent out-of-set faces were both used to assign a

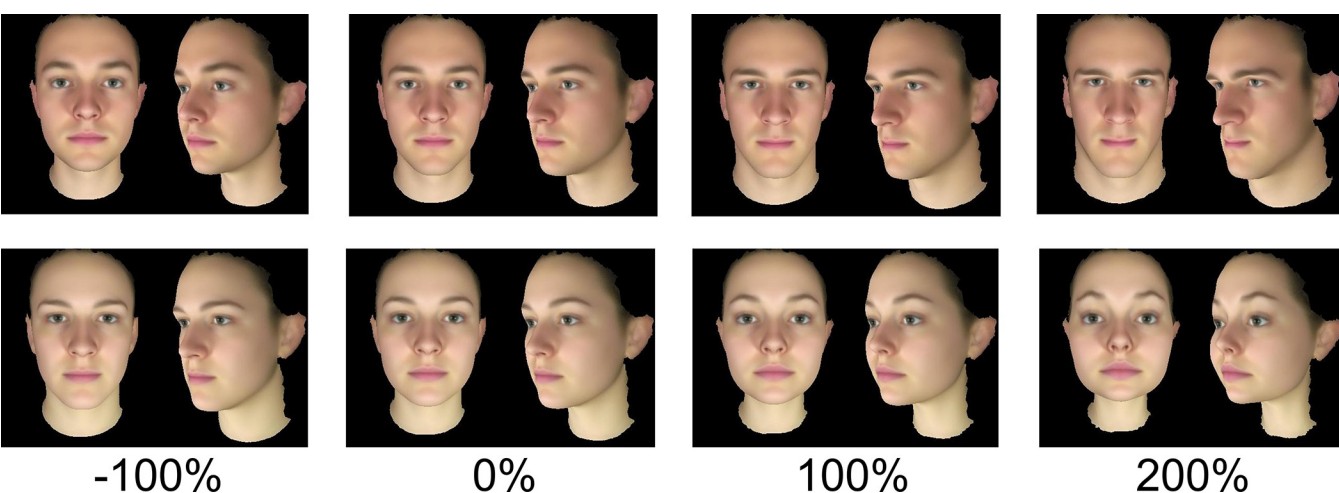

**Fig 1. Transformation of sexually dimorphic face shape.** Upper: The average male face transformed by −100% (feminised), 0% (neutral), +100% (masculinised) and +200% (masculinised) sexual dimorphism. Lower: The average female face transformed by −100% (masculinised), 0% (neutral), +100% (feminised) and +200% (feminised) sexual dimorphism. To preserve anonymity of participants, these images are averages of the shape, colour and texture of 50 male faces and 68 female faces, respectively, rather than of real individuals.

morphological masculinity score [49,50] for each of the 118 faces. The two sets of scores correlated (N = 118, Pearson's r = .78) [51]. This shows that the face sexual dimorphism shape vector has external validity.

To provide more information on the shape transformation applied [49], shape dimorphism score varied within the original sample of 68 female faces SD = 33% and within the original sample of 50 male faces SD = 36%. The difference in facial shape dimorphism scores between the original sample of male and female faces had an effect size Cohen's D = 0.34. The two male base faces were each feminised and masculinised in shape within a range from −100% to +200%, and the two female base faces were each masculinised and feminised in shape within a range from −100% to +300% sexual dimorphism. For both female and male faces, 0% corresponds to the original, unmanipulated base face. For male faces, negative values correspond to a decrease and positive values to an increase in masculinity. For female faces, negative values correspond to a decrease and positive values to an increase in femininity. A less extreme level of manipulation was chosen for male faces because exaggeration beyond 200% caused glitches which detracted from the appearance.

The 3D stimuli were imaged in two poses (a frontal and a half profile view). Two views were simultaneously presented to provide more 3D information (e.g., about chin shape and eyebrow structure) and because different views affect attributions differently [52]. Participants were able to transform the faces in increments of 10%, hence there were 30 increments for male faces and 40 for female.

## Procedure

Participants were asked to complete a demographic questionnaire about age, gender, sexual orientation. They were then presented with a series of two male and two female face stimuli with sliders below each stimulus that allowed them to manipulate the level of sexual dimorphism in the display. Participants were not told of the underlying nature of the shape transform.

Instructions were given before each set of two faces, requiring the participants to use their computer mouse to move the slider and adjust the face to best match the task description.

Participants were asked to transform same-sex faces to represent their own face (the instructions were "alter the male/female face to make it closest in appearance to your own face"), and the same for their ideal face. They were also asked what they thought the opposite sex would find attractive in a face for a short-term and long-term relationship. An example instruction was "alter the male face to make it most attractive to heterosexual women for a short-term (sexual) relationship". The two transformed base faces were presented in random order for each instruction, before progressing to the next instruction. The order of long- and short-term judgements was randomised. Then, participants were asked to transform opposite-sex faces to represent what they found most attractive for a short-term sexual and long-term relationship. There was no time limit for trials; participants proceeded to the next stimulus when they had completed their choice. The starting position of the slider (and therefore, the degree of feminisation/masculinisation) was randomly assigned on each trial.

Participants' selection of sexual dimorphism of the two faces was correlated for each judgment. For the 72 men, judgments for the 2 example faces correlated for own facial masculinity (r = .67), ideal facial masculinity (r = .82), prediction of male facial masculinity preferred by women for short- (r = .72) and long-term relationships (r = .69) and preference for female facial femininity for short- (r = .59) and long-term (r = .66) relationships. Likewise for the 72 women, judgments for the 2 example faces correlated for own facial femininity (r = .72), ideal facial femininity (r = .60), prediction of female facial femininity preferred by men for short- (r = .70) and long-term relationships (r = .63) and preference for male facial masculinity for short- (r = .69) and long-term (r = .72) relationships. Given the consistent and high correlation, the values for the 2 example faces for each judgment were averaged together and the average values entered as dependent variables for analysis in SPSS.

## Statistical analyses

Data analysis was carried out using SPSS 26.0. For each trial type, the average level of sexual dimorphism selected across the two base faces was computed. Choices are expressed as percentages, where 0% represents the unmanipulated base face, positive percentages represent transformations of increased sexual dimorphism and negative percentages represent transformations of decreased sexual dimorphism (masculinity for males and femininity for females).

Data from men and women were checked separately for outliers. Data values above or below 3 standard deviations from the mean were eliminated to avoid undue leverage. This eliminated three values. Skew and kurtosis of data in each task were acceptable [53]. Therefore, parametric tests were used.

Data were analysed in a 2-way mixed ANOVA with level of sexual dimorphism as the dependent variable, gender of participant (2 levels: men, women) as a between-subjects variable and relationship context (2 levels: short-and long-term) as a within-subjects variable. Separate models were used to analyse preferences for male and female faces.

## Results

### Male faces: Masculinity predictions and preferences

A two-way mixed ANOVA for male facial masculinity revealed a main effect of participant gender ($F_{[1,141]}$ = 28.26, p < .001, $\eta_p^2$ = .167). Women preferred a lower level of masculinity in male faces (Mean = 32.7%, SE = 5.8%) compared to what men predicted women would prefer (Mean = 76.5%, SE = 5.8%). Hence, men exaggerated the level of masculinity preferred by women, supporting hypothesis H1.

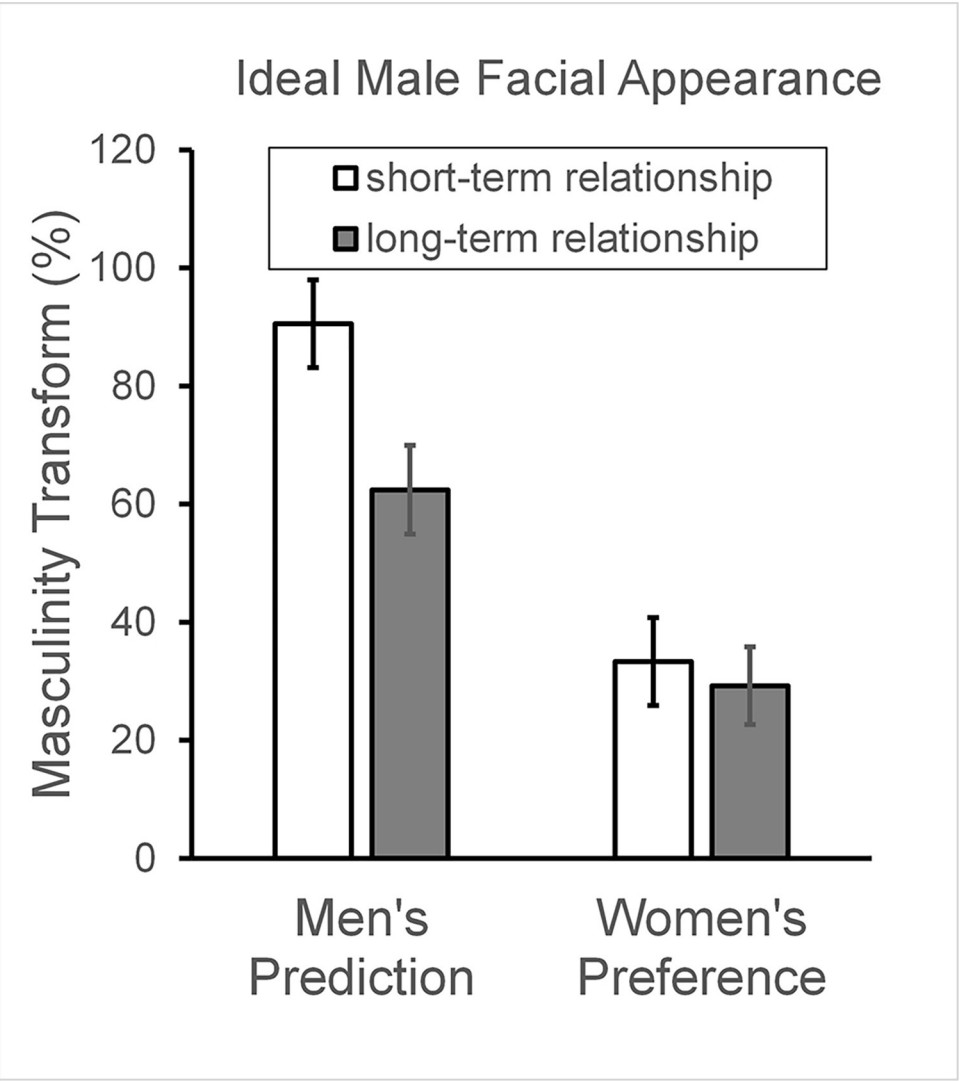

**Fig 2. Male facial masculinity.** Left: Men's predictions of women's preferences for short- and long-term relationships. Right: Women's masculinity preference for short- and long-term relationships. Histogram bars give the mean and SE of the percentage masculinity chosen, with 0% corresponding to the original base face.

There was a significant main effect of relationship context ($F_{[1,141]}$ = 7.95, p = .005, $\eta_p^2$ = .053) with a higher level of masculinity chosen for a short-term relationship (Mean = 62.34%, SE = 5.3%) than a long-term relationship (Mean 46.8%, SE = 4.6%),

The effect of relationship context was qualified by an interaction with participant gender ($F_{[1,141]}$ = 5.16, p = .025, $\eta_p^2$ = .035, see Figs 2 and 3). Men perceived a difference in masculinity preferred by women in short- vs long-term contexts (Fig 2 left) but women showed no such difference in preference (Fig 2 right). That is, men thought women would prefer a higher level of masculinity for short-term relationships (Mean = 90.6%, SE = 7.4%) than for long-term relationships (Mean = 62.4%, SE = 6.5%, Bonferroni corrected comparison p < .001). By contrast, women showed similar preferences for level of masculinity in short-term (Mean = 34.2%, SE = 7.5%) and long-term (Mean = 31.2%, SE = 6.5%, p = .700) relationships. Men's greater misperception in short-term than long-term relationship supports H2. For a visualisation of the facial shapes chosen by men and women in the different relationship contexts see Fig 3.

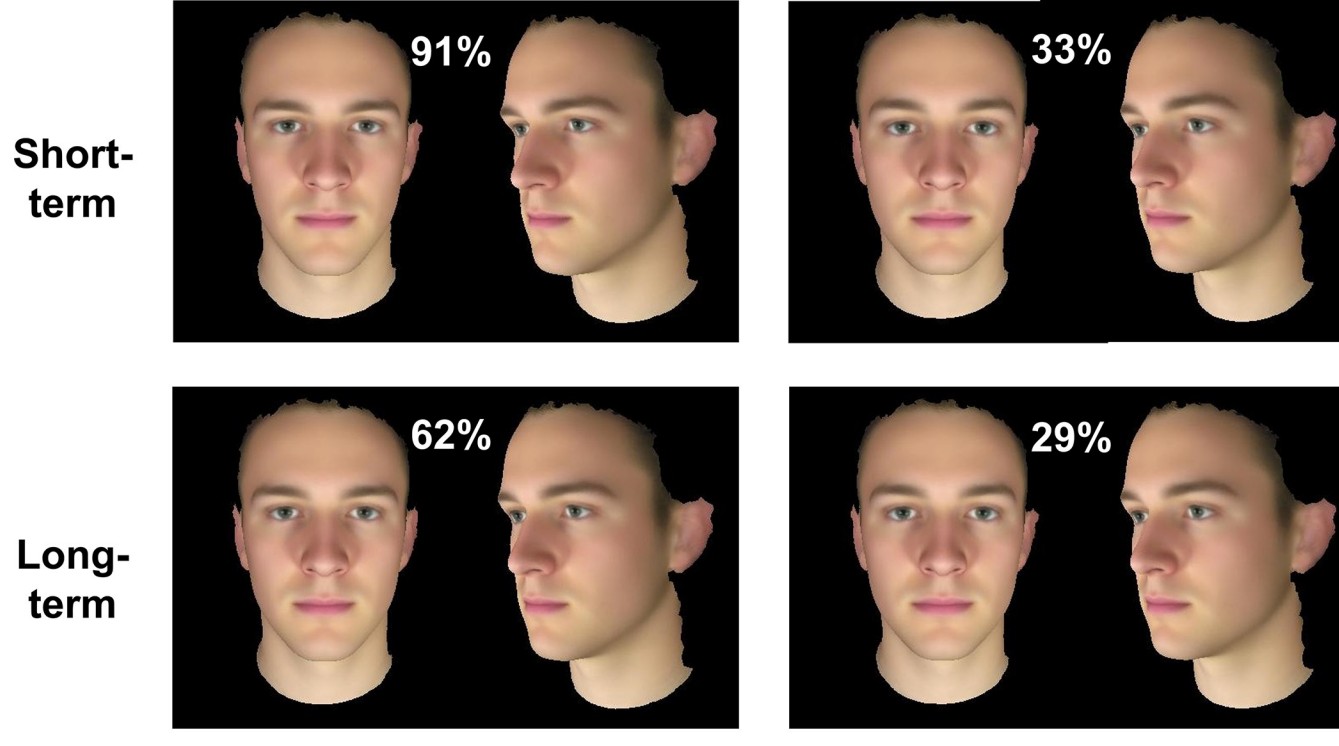

**Fig 3. Male facial masculinity chosen.** Men's prediction of women's preferences (left) and women's masculinity preference (right) for short-term and long-term relationship contexts. Numbers in white display the mean amount of sexual dimorphism transform applied. The facial images are averages of the shape, colour and texture of 50 male faces and 68 female faces, respectively, rather than of real individuals.

### Female faces: Femininity preferences and predictions

A similar 2-way mixed model ANOVA was conducted on the data for female face choice. The ANOVA for female facial femininity revealed a main effect of participant gender ($F_{[1,141]}$ = 34.5, p < .001, $\eta_p^2$ = .197, Fig 4). Men preferred a lower level of femininity (Mean = 124.8%, SE = 5.7%) compared to what women predicted (Mean = 172.1%, SE = 5.7%), although both men and women selected high levels of sexual dimorphism in female faces as being desirable. This confirms hypothesis H3 for female facial appearance.

There was a significant main effect of relationship context ($F_{[1,141]}$ = 5.4, p = .021, $\eta_p^2$ = .037, Fig 4), reflecting a higher level of femininity being desired for a short-term relationship (Mean = 155.3%, SE = 5.3%) compared to a long-term relationship (Mean = 141.6%, SE = 4.6%).

The interaction between participant gender and relationship term was non-significant ($F_{[1,141]}$ = 2.7, p = .106, $\eta_p^2$ = .018). The non-significant interaction does not support Hypothesis H4. For a visualisation of the female facial shapes chosen by men and women in the two relationship contexts see Fig 5.

### Own and ideal facial dimorphism

Both men and women reported that their ideal level of sexual dimorphism was higher than their own level of sexual dimorphism. On average, men estimated their own level of facial masculinity as 37.2% (SE = 6.5%) and their ideal level of facial masculinity as 65.3% (SE = 7.2%),

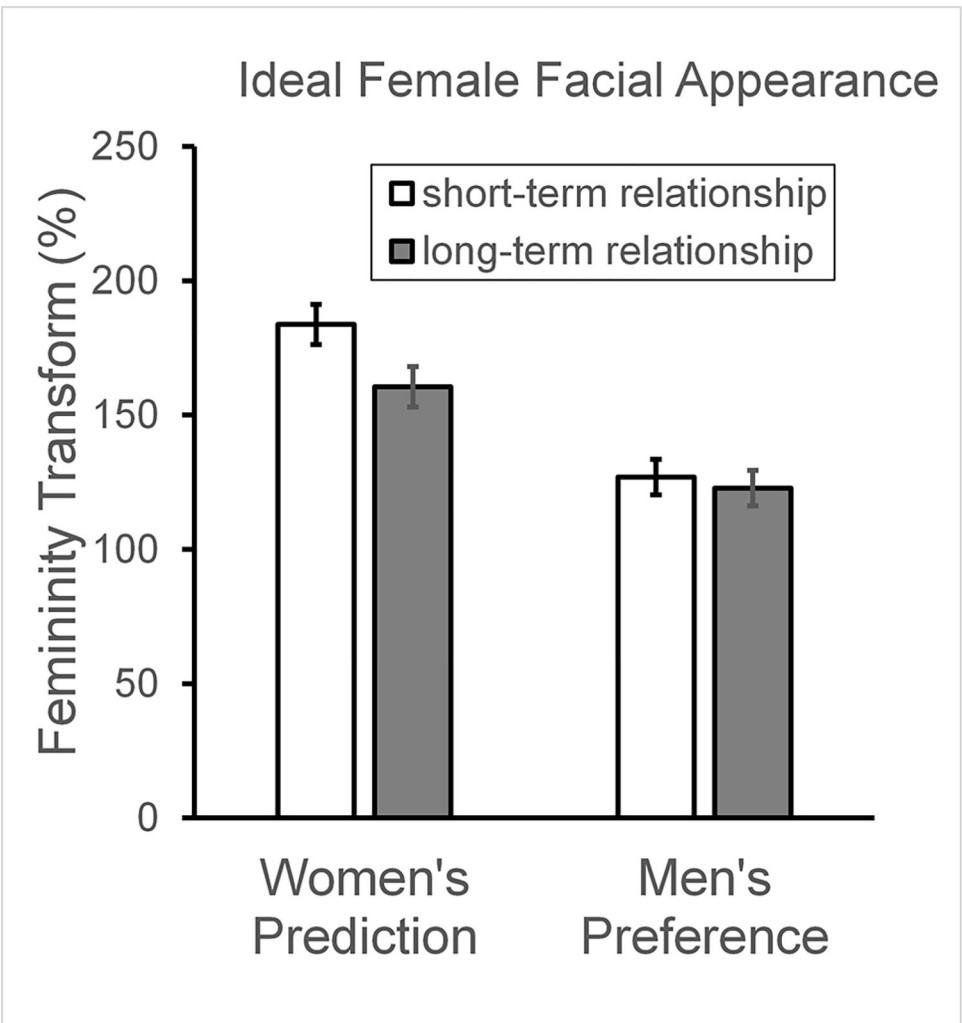

**Fig 4. Female facial femininity.** Left: women's predictions of men's preferences for short- and long-term relationships. Right: Men's masculinity preference for short- and long-term relationships. Histogram bars give the mean and SE percentage masculinity chosen, with 0% corresponding to the original base face.

which is significantly higher (t(71) = 3.69, p < .001; Cohen's D = 6.5). Similarly, on average women estimated their own level of facial femininity as 124.0% (SE = 8.1%) and their ideal level of femininity as 161.5% (SE = 6.2%), levels which differed significantly (t(71) = 4.29, p < .001; Cohen's D = 7.4). These findings support hypothesis H5.

### Dissatisfaction with facial dimorphism

Following studies of body image dissatisfaction, we computed an index of facial appearance dissatisfaction defined as ideal face dimorphism minus own facial dimorphism. For men positive values of the index expresses desire to have a more masculine face, and for women positive values of the index expresses a desire to have a more feminine face.

For participant women, ANCOVA (analysis of covariance) with prediction of men's desires of female facial femininity for short- and long-term as a repeated measure and facial dissatisfaction as a covariate showed a main effect of facial dissatisfaction ($F_{[1,69]} = 6.12$, p = .016, $\eta_p^2$ = .081). The greater the facial femininity women predicted men desired, the higher the

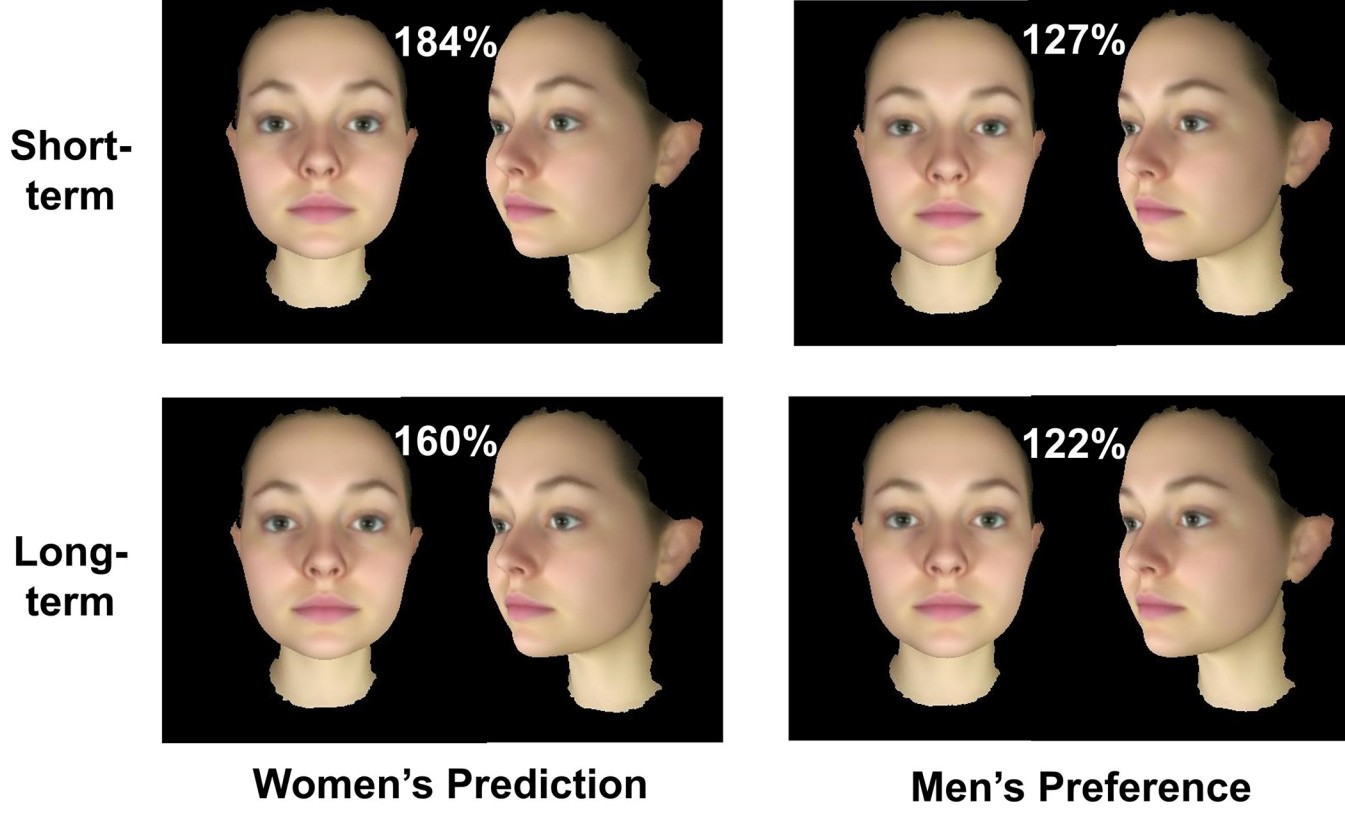

**Fig 5. Female facial femininity chosen.** (Left) Women's prediction of men's preferences for short-term and long-term relationship contexts. (Right) Men's femininity preference for short-term and long-term relationship contexts. The facial images are averages of the shape, colour and texture of 50 male faces and 68 female faces, respectively, rather than of real individuals.

women's facial dissatisfaction was. There was no main effect of relationship term ($F_{[1,69]}$ = 2.57, p = .113, $\eta_p^2$ = .036) and no interaction between relationship term and the facial dissatisfaction index ($F_{[1,69]}$ = 2.29, p = .135, $\eta_p^2$ = .032).

For participant men, ANCOVA was conducted with prediction of women's desires of male facial masculinity for short- and long-term relationships as a repeated measure and facial dissatisfaction as a covariate. This analysis showed a main effect of facial dissatisfaction ($F_{[1,69]}$ = 5.41, p < .023, $\eta_p^2$ = .073). The greater the facial masculinity men predicted women desired, the higher the men's facial dissatisfaction score was. There was a main effect of relationship term ($F_{[1,69]}$ = 6.23, p = .015, $\eta_p^2$ = .083) reflecting men's prediction that women preferred higher levels of facial masculinity for short- rather than long-term relationships. There was no interaction between relationship term and the facial dissatisfaction index ($F_{[1,69]}$ = 0.08, p = .775, $\eta_p^2$ = .001)."

Equivalent ANCOVAs with ideal dimorphism as a covariate show that it is strongly related to the misperception of opposite-sex desires (see Supporting Information). Mediation analysis included in the Supporting Information confirmed the mediation role of ideal dimorphism in the relation between misperception and dissatisfaction.

## Discussion

### Summary of results

We used interactive 3D head models to allow participants to adjust the sexual dimorphism of male and female faces. We asked men and women for their predictions of the face shape that

the opposite sex find ideal for a short- or long-term relationship. We measured ground truth by asking participants what face shape they found attractive in a short- or long-term partner. Comparing these two measures revealed that predictions made by men and women of opposite-sex desires were exaggerated. Men predicted that women desired greater levels of facial masculinity than women actually desired. Reciprocally women predicted men desired higher levels of facial femininity than men desired. These results support hypotheses H1 and H3. Predictions for opposite-sex desires were exaggerated for both long-term and short-term relationships. Men's misperception of desirable levels of masculinity was more pronounced for short-term relationships than long-term relationships, as predicted (in line with H2). Women's misperception, in contrast, was independent of relationship context (i.e., not in support of H4).

Additionally, each participant adjusted same-sex models to approximate their own appearance and their ideal appearance. These two measures revealed that most men and most women were not content with their facial appearance and wanted an increase in the sexual dimorphism of their face shape, consistent with hypotheses H5 and H6. For both men and women the magnitude of dissatisfaction with face shape was predicted by the degree that participants exaggerated the facial appearance that the other sex desired, as predicted by hypothesis H7.

## Exaggeration in predictions of opposite-sex desires

The main finding here was that participants consistently overestimate the degree of sexual dimorphism in face shape that the opposite sex desires. Exaggeration in terms of predictions for opposite-sex desires has been seen frequently in research on body shape ideals. Men overestimate the muscularity and weight that women desire [9,11–14]. Reciprocally, women overestimate the thinness that men desire [9–12]. What is novel here is that the stimuli we used were model heads and the dimension manipulated was sexual dimorphism in face shape.

As argued in the Introduction, thinness in body morphology is stereotypically associated with femininity [15–17] in women and muscularity is stereotypically associated with masculinity in men [15,18,19]. The misperception of opposite-sex desires for both body and face shape may in part reflect a misperception of sexual dimorphism desired.

Participants adjusted face shape here but were never told that the adjustment affected masculinity/femininity in facial morphology. With feminisation, the female facial stimuli become more rounded, suggesting an increase in adipose tissue in the cheek area. It is therefore surprising that women choose higher levels of facial feminisation since women usually express a desire for lower levels of body fat. It is likely that it is the distribution of fat within the face and body that is important, not just the amount of fat. For example, for women, adipose tissue in the buttocks and breast regions is more desirable than in the abdominal region. As masculinity in face shape increased, neck girth also increased. Neck girth and shape could act as cues to overall weight and to muscularity. Changing the level of sexual dimorphism affects every feature of the face not just the cheeks or neck, and we presently do not know which facial feature or features drive preferences.

## Context of attractiveness judgements

When relationship context is not specified (e.g., [54]), exaggeration of predictions of opposite-sex views may reflect, in part, different interpretations of the task. For example, women might think that men are judging female facial attractiveness in the context of a short-term relationship, while men might judge female facial attractiveness for a long-term relationship. Here, judgements of face shape were made separately for both a short-term sexual relationship and a long-term partnership. Therefore, misperception of opposite-sex views evident here cannot

reflect differences in the interpretation of relationship type. Indeed, exaggeration of dimorphism preferred by the opposite sex occurred in both short- and long-term relationship contexts.

From the perspective of intra-sexual competition, attempting to form a durable partnership between individuals with very discrepant attractiveness is not a secure strategy as the more attractive partner may be poached by a competitor of equal attractiveness. When considering a short-term relationship there is no long-term commitment made by either party, hence mate guarding is not a worry. In this short-term context, the participants are not constrained in their desires by pragmatic considerations of long-term relationship stability. Hence, participants can express and act on more extreme preferences.

### Dissatisfaction with facial appearance

A surprising finding of the study was that most participants (61% of men, and 73% of women) indicated that their own facial dimorphism was less than their ideal. These figures are comparable to dissatisfaction with body shape with 75% of girls aged 16–18 [55] and 66% of women wanting to be lighter [56]. For men, about the same proportion want to gain body weight as want to lose weight [55,57]. Indeed, Cachelin et al. [56] report 26% men want to gain weight. Thus, the proportion of men wanting to increase facial masculinity is greater than the proportion of men wanting to increase body weight. Body image dissatisfaction in men incorporates a desire for a lean and more muscular torso, and a larger proportion (90%) of US undergraduate men would like to be more muscular [58].

Lei and Perrett [12] found that misperception of what the other sex desires in terms of body shape predicted body image dissatisfaction (defined as own minus ideal body shape). The more extreme the misperception of the desire by the opposite sex, the greater the dissatisfaction with own body. This relationship between body image dissatisfaction and misperception of opposite-sex desire was apparent in both men and women. The results here indicate an analogous effect for face shape.

Dissatisfaction with own facial dimorphism was predicted by misperceptions of opposite-sex preferences in both men and women. Specifically, for men, the more they thought that women preferred high facial masculinity for short-term and long-term partners, the more dissatisfied the men were with their own face shape. Likewise, for women, the more they thought that men preferred high levels of facial femininity for short-term and long-term partners, the more they were to be dissatisfied with their own facial femininity. Analysis in Supporting Information suggested that the relationship between misperception of opposite-sex desires and facial dissatisfaction was mediated by ideal dimorphism. Misperception of desires was directly associated with exaggerated ideals and exaggerated ideals were in turn directly associated with a higher facial dissatisfaction score. The direct pathway between misperception and dissatisfaction was non-significant when both ideals and misperception were used as predictors of dissatisfaction.

The results suggest that misperceptions of opposite-sex preferences for face shape affect an individual's satisfaction with their facial shape. Thus, these findings parallel those on body shape, where misperception of the opposite sex's preference for body shape was associated with a high score on a body image dissatisfaction index [12,59].

The high ideal level of dimorphism chosen by participants is potentially worrying. Women and men may attempt to alter their appearance to conform to an unrealistic ideal. Indeed, cosmetic surgery is used to enhance aspects of female facial dimorphism such as lip plumpness [60] and arched shape of eyebrows [61]. Likewise, men striving for a more masculine appearance may use supplements, including steroids [62].

## Preferences for high levels of sexual dimorphism

One unexpected aspect of the results was the very high level of exaggeration that was preferred. Part of the exaggeration of facial shape may derive from features typically present in real faces that were absent in the stimuli. The 3D faces were cropped to remove hair; this made them unnaturally bald. For female faces this baldness may contribute to an increase of masculinity, hence, to reset a normal level of femininity participants may have increased the femininity of the shape of other facial features.

Previous work has found that femininity in female faces is attractive [22,63]. Male facial masculinity on the other hand has been found to be more variable in its effect on attractiveness [22]. The levels of sexual dimorphism for optimal attractiveness found here are higher than commonly reported although they do align with measures taken with the same type of 3D face stimuli [28]. This may be due to the presentation of face and profile views which can elicit different judgements [52]. Alternatively, some of the textural details that contribute to gender recognition (e.g., a 5 o'clock shadow from a shaved beard may be lost in the formation of stimuli, hence shape cues need amplifying to compensate. For example, female and male skin texture may differ in coarseness; by creating composite textures, some of this coarseness of male faces would have been lost, making them appear more feminine. Another explanation could be the use of ranges that were asymmetrical (for male faces, this was −100% to +200% and for female faces −100% to +300%). This might have briefly shifted participants' perception of what is a "normal" level of sexual dimorphism and hence might have artificially inflated preferred levels of dimorphism (e.g., [64] particularly for female faces).

That said, for the current study what is crucial is not the absolute but the relative levels of dimorphism, for example men's preferences vs women's predictions of men's preferences. Similarly, it is the relative level of dimorphism between own and ideal face shape that defines the dissatisfaction with own facial shape.

## Limitations

One limitation is that we used an index (own minus ideal facial dimorphism) as a measure of dissatisfaction with facial appearance. It is possible that individuals can have a high score on the index used but may not be unduly troubled by their facial appearance. Thus, in the future, it would be appropriate to compare the index of dissatisfaction with self-reports of the degree of dissatisfaction with face shape. Alternatively, one might ask participants if they would consider cosmetic procedures (such as collagen injections) that are designed to enhance facial femininity, or appearance-enhancing supplements that might affect muscularity. These self-report measures could be used to estimate the validity of the index of facial appearance dissatisfaction. Our study involved White models (from diverse populations within Europe) and White participants currently resident in the UK. Future studies could explore population differences within and outside Europe extending to a range of ethnic groups, as body dissatisfaction shows consistent (but relatively small, d = 0.3) differences between White and Black women [65]. An improvement to the generalisability of the study would be to use sexual dimorphism from more than one set of faces.

Choices were made within the context of specific instructions (e.g. "alter the male face to make it most attractive to heterosexual women for a short-term (sexual) relationship"). The instruction does not specify the population of women for whom the participant is to predict preference. Future research could target different populations distinguishing between peer group and a younger or more attractive group.

Most research on facial appearance has been performed with frontal images of the face. The profile view of the head provides additional information about the neck, chin and cheek

structure which may be important for attributions. Indeed, the profile pose of male faces decreases attractiveness and increases apparent dominance [52]. By including both face and half-profile views our results may have greater generality than studies using a single view. Future work testing the effects of the face, or the profile view alone could confirm the viewpoint specificity or generality of masculinity (mis)perceptions. Partner's status has been found to affect facial attractiveness judgments [66,67] so could be included as an additional factor in future studies.

We cannot assume complete accuracy in participant's judgments about their own faces. Research using 3D scans of participants' faces could define objective levels of sexual dimorphism in face shape. Nonetheless, subjective self-assessment is important as viewing attractive or unattractive others changes self-esteem and modifies face and body ideals [68,69].

## Conclusion

We have demonstrated a substantial misperception of what men and women predict the opposite sex to desire in terms of sexual dimorphism of face shape. Men overestimate the masculinity women desire, and women overestimate the femininity men desire. Our study also reveals widespread dissatisfaction with facial appearance. Most men and women had an ideal level of facial dimorphism that was greater than their own facial dimorphism. The dissatisfaction and misperception are related, in that the greater the misperception of others' desires, the greater the index of dissatisfaction. These findings parallel the misperception that has been documented for desires about body shape. The results suggest that misperception of others' desires and dissatisfaction with self-image are not restricted to body thinness and muscularity but include the dimension of femininity-masculinity.

Research on appearance dissatisfaction has focused on body shape and composition. While there are cues to body weight and proportion of body fat evident in face shape [49], our research indicates that a common appearance concern reflects sex-typicality in facial appearance. Indeed, distorted ideals of body weight and muscularity may reflect underlying concerns about sex-typicality. Appearance dissatisfaction is an important issue as it contributes to mental and physical health. Our findings, therefore benefit the scientific literature and public knowledge by providing insight into the nature of harmful appearance ideals. Future research exploring strategies to moderate aberrant notions of (and attention to) sex-typicality in body or face shape may have benefits to self-esteem and health.

## Supporting information

**S1 Fig. Predictors of facial dissatisfaction.** Linear regression models of the direct and indirect effects of facial dimorphism predicted to be desired by the opposite sex for participant women (upper) and men (lower). The flow chart shows the standardised independent direct effects (β values, * p < 0.05) of predicted dimorphism desired by the opposite sex on ideal dimorphism and facial dissatisfaction.
(JPG)

**S1 File.**
(DOCX)

## Acknowledgments

We give special thanks to Sabrina Hallier for discussion of results, Anne Perrett for proofreading the manuscript and to Dengke Xiao for helping with the software.

## Author Contributions

**Conceptualization:** David I. Perrett, Iris J. Holzleitner, Xue Lei.

**Formal analysis:** David I. Perrett.

**Methodology:** David I. Perrett, Iris J. Holzleitner.

**Writing – original draft:** David I. Perrett.

**Writing – review & editing:** David I. Perrett, Iris J. Holzleitner, Xue Lei.

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
