## [Decision Letter · Decision Letter 0]

10 Jul 2024

PONE-D-24-22275Misperception of the facial appearance that the opposite sex desiresPLOS ONE

Dear Dr. Perrett,

Thank you for submitting your manuscript to PLOS ONE. After careful consideration, we feel that it has merit but does not fully meet PLOS ONE’s publication criteria as it currently stands. Therefore, we invite you to submit a revised version of the manuscript that addresses the points raised during the review process.

Both reviews, for which I am grateful, state that your paper makes an important contribution and emphasize that representativeness is an issue that needs addressing, which agrees with my own reading. Please can I ask you to carefully address all points raised, with particular emphasis on representativeness. In this context you might find Bovet, Tognetti & Pollet (2022) helpful.

I would also like you to address the following points:

Please state clearly if you pre-registered your hypotheses and, if applicable, where the pre-registration can be found.Please justify your sample size (e.g., Lakens, 2022, Collabra).L76, “good quality”: please explain.L129, “participants’ estimate of their ideal”: Why is this an estimate? Reconsider wording.L138, “Long-term relationships come with direct benefits of parental investment”: Benefit for whom?L153, “we will”: reconsider tense.L203, “composite stimuli … more representative”: Of what? Reference needed.L209, “with the distance from the average female face to the average male face being equivalent to a +100% increase”: Please indicate SDs within female and male stimuli using this measure. (In other words, what is the effect size for the female/male difference in the stimuli?)L213 and L223 seem to contradict each other.L232, “presented with a series of male and female face”: How many? If there was more than one stimulus for each DV, please provide some information on reliability/internal consistency.L336: I don’t see how perceived level of own facial masculinity is a sensible predictor in your regression, given that it was already used to compute your criterion.L499, “involved White models and predominantly White UK participants”: How does this agree with your “White ethnicity” pre-screening criterion (L191)? ==============================

We look forward to receiving your revised manuscript.

Kind regards,

Johannes Hönekopp

Academic Editor

PLOS ONE

Journal Requirements:

3. We note that Figures 1, 3 and 5 includes an image of a participant in the study. 

Reviewers' comments:

Reviewer's Responses to Questions

**Comments to the Author**

1. Is the manuscript technically sound, and do the data support the conclusions?

Reviewer #1: Yes

Reviewer #2: Yes

2. Has the statistical analysis been performed appropriately and rigorously? 

Reviewer #1: Yes

Reviewer #2: Yes

3. Have the authors made all data underlying the findings in their manuscript fully available?

Reviewer #1: Yes

Reviewer #2: Yes

4. Is the manuscript presented in an intelligible fashion and written in standard English?

Reviewer #1: Yes

Reviewer #2: Yes

5. Review Comments to the Author

**Reviewer #1: **The manuscript presents sound data with clear findings. Hypothesis set are derived from literature and unsurprisingly met in the findings. The stimuli is appropriate, though I question whether it is useful to include both front facing and three-quarter facing view within the same stimulus. Typically the face is viewed from the front and whilst body fat is easiest to see from three quarter view, for facial attractiveness, I believe, the front view would be sufficient. Please could further justification and discussion be added to address this concern. Furthermore, the stimulus is presenting Caucasian white male/female. It needs to be acknowledged that even within Caucasians, the facial features vary depending on which exact country a person originates from. Whist UK residency was an inclusion criteria, this cohort might include Scandinavian, Eastern European etc. participants whose facial features are very different to typical British facial features. Furthermore, how was it assured that the stimuli was presenting typical British facial features. Further discussion and justification required.

Data is analysed appropriately using a set of ANOVAs. It is though not clear why the participants' own relationship status is not included in the analysis. I believe this likely to have a contributing effect. Please discuss.

My main concern is though 'so what'. Why is this important. Please could the authors add contemplation of why the current findings are important and how the line of research benefits the literature and general public knowledge. Clear indications of future research need to be equally identified.

**Reviewer #2:** This was a well written manuscript with straightforward findings: both heterosexual men and women believe that the opposite sex prefers higher level of sexual dimorphism.

I would like to see a bit more discussion of the following points:

1. The scale for male and female stimuli was different, because in the case of male stimuli the highest exaggeration resulted in artifacts. But participants would generally adapt to the range of the scale and use it accordingly. If this is not accounted for, the results are a bit strange. First, there is about 100% difference between the responses for male and female faces. Second, this is also the case for self-assessment of facial shapes (37.2% for men vs. 124% for women). These results either simply reflect the very different ranges of the scales or strong bias in the case of female faces (or alternatively, very inaccurate perceptions).

2. Participants are making decisions about their own face shape. But how can we know that they are accurate at all?

3. This is related to point 2 and the regression analyses. Relying on difference scores is tricky. It would be nice to see some raw data/ means before the differences are computed. A participant with 10% "self" shape and 50% "ideal" is different from one with 10% and 70% and possibly one with 30% and 70%.

4. One conceptual point is that the main finding is based on the difference between predictions and preferences. But in principle, we don't know what population participants are having in mind when making their predictions. Another way to put it is that one needs to assume that samples are representative of the population of interest. At the very least, this needs to be discussed under limitations.

6. PLOS authors have the option to publish the peer review history of their article (what does this mean?). If published, this will include your full peer review and any attached files.

Reviewer #1: No

Reviewer #2: No

---

## [Author Response · Author response to Decision Letter 0]

18 Aug 2024

each point made by the editor and the 2 reviewers has been addressed in the document "Response to Reviewers" up loaded in attached files

Both reviews, for which I am grateful, state that your paper makes an important contribution and emphasize that representativeness is an issue that needs addressing, which agrees with my own reading. Please can I ask you to carefully address all points raised, with particular emphasis on representativeness. In this context you might find Bovet, Tognetti & Pollet (2022) helpful.

Response. Bovet et al. (2022) draw attention to the limited size of samples used to create some prototypes. Holtzman (2018) creates a prototype of high psychopathy from a small subset of 33 males. Indeed the same set of male faces contributed to prototypes for 47 behavioural traits. Bovet et al. (2022) make valid criticisms of prototype limitations. 

We used a large set of faces (n = 118) to construct the sex dimorphism vector. Furthermore, we tested the external validity of the sexual dimorphism vector. This was done by assigning each of the 118 faces a morphological masculinity score derived from the within-set sexual dimorphism vector [49]. Additionally, a second out-of-set sexual dimorphism vector was derived from an independent set of 40 male and 40 female faces [50]. This out-of-set dimorphism vector was used to define a second set of morphological masculinity scores for the 118 faces. The morphological masculinity scores from the within-set and the out-of-set sexual dimorphism vectors were correlated (N = 118, Pearson’s r = .666). 

This is described in text: “Thus, each of the 118 faces contributed to the definition of the dimorphism vector along which base faces were manipulated. Importantly, we compared the sexual dimorphism from a second independent set of 40 male and 40 female faces [50]. The sexual dimorphism vectors from the within-set and the independent out-of-set faces were both used to assign a morphological masculinity score [49,50] for each of the 118 faces. The two sets of scores correlated (N = 118, Pearsons r = .666). This shows that the face sexual dimorphism shape vector has external validity.”

We acknowledge the limitations in text “An improvement to the generalisability of the study would be to use sexual dimorphism from more than one set of faces.”

I would also like you to address the following points:

• Please state clearly if you pre-registered your hypotheses and, if applicable, where the pre-registration can be found.

We now state that the hypotheses were not pre-registered. “The study hypotheses were not pre-registered.”

• Please justify your sample size (e.g., Lakens, 2022, Collabra).

The methods now include sample size considerations. “Lei and Perrett [12] reported misperception of body shape (body mass index) preference between men and women with an effect size ranging between d = 0.4 and d = 0.9. If we assume that misperception of facial shape preferences has a similar range in effect size we can estimate an appropriate sample size for difference between two independent means (samples). With a power of 80%, an alpha error of 0.05 and an effect size of d = 0.4, G-power suggests a sample size of 76 men and 76 women.”

• L76, “good quality”: please explain.

Quality here is defined in the specialised and limited sense of competition in the ‘mating market’ [25]. We have added: “Good quality from this limited perspective refers to individuals that are desired as a partner, and who can out-compete rivals in their own quest to secure a desirable partner. Multiple attributes contribute to mate quality (including resources, youth and physical attractiveness).” 

We conclude the paragraph by noting: “Hence, facial masculinity in men can be considered as a sign of quality in the mating market”.

• L129, “participants’ estimate of their ideal”: Why is this an estimate? Reconsider wording. 

The sentence has now been simplified and the term “estimate” deleted: “This index can be computed as the difference between the participant’s ideal facial dimorphism and the participant’s own facial dimorphism (ideal minus own dimorphism).” 

• L138, “Long-term relationships come with direct benefits of parental investment”: Benefit for whom?

The recipient of direct benefits would be the participant expressing mate preferences. This is now explicitly stated: “Long-term relationships come with direct benefits (to the participant) of parental investment, resources and safety, whereas short-term relationships can have indirect genetic benefits (to the participant’s offspring) but only temporary resources [44].”

• L153, “we will”: reconsider tense.

“Will” is deleted: “To test this, we measure the male face shape that participant women desire and compare this to the face shape that men predict women will desire. Reciprocally, we measure the female face shape that participant men desire.”

• L203, “composite stimuli … more representative”: Of what? Reference needed.

Composite facial stimuli have a long history. Composites made by combining several facial images average out idiosyncratic or inconsistent features and maintain features that are common to a group. We now explain and cite a review of composite facial stimuli.

“Composite stimuli were employed because they are more representative (of the category of faces from which they were constructed) than the original faces [48]. Additionally, the composite images cannot be recognised as known individuals.”

“[48]. Perrett DI. Representations of facial expressions since Darwin. Evolutionary Human Sciences, 2022;4, e22 doi: 10.1017/ehs.2022.10”

• L209, “with the distance from the average female face to the average male face being equivalent to a +100% increase”: Please indicate SDs within female and male stimuli using this measure. (In other words, what is the effect size for the female/male difference in the stimuli?)

The following text has been added to answer the question about SD and effect size. “To provide more information on the shape transformation applied [49], shape dimorphism score varied within the original sample of 68 female faces SD = 36.5% and within the original sample of 50 male faces SD = 42.8%. The difference in facial shape dimorphism scores between the original sample of male and female faces had an effect size Cohen’s D = 0.72.”

• L213 and L223 seem to contradict each other.

Line 213 refers to the ‘base’ faces which were subsequently transformed in sexually dimorphic shape (as described in line 223) to create a range in appearance that the participants chose. 

This has been clarified in text: “The study used 3D stimuli. The stimuli were two male and two female ‘base’ faces. Each of the male base faces was a composite or average of the shape, colour and texture of three male faces. Likewise, each of the female base faces was a composite of shape, colour and texture of three female faces.

And 

“The two male base faces were each feminised and masculinised in shape within a range from −100% to +200%, and the two female base faces were each masculinised and feminised in shape within a range from −100% to +300% sexual dimorphism.”

• L232, “presented with a series of male and female face”: How many? If there was more than one stimulus for each DV, please provide some information on reliability/internal consistency.

There were 2 male face stimuli, or 2 female face stimuli used for each facial judgment. Measures of consistency have now been added. “Participants’ selection of sexual dimorphism of the two faces was correlated for each judgment. For the 72 men, judgments for the 2 example faces correlated for own facial masculinity (r = .67), ideal facial masculinity (r = .82), prediction of male facial masculinity preferred by women for short- (r = .72) and long-term relationships (r = .69) and preference for female facial femininity for short- (r = .59) and long-term (r = .66) relationships. Likewise for the 72 women, judgments for the 2 example faces correlated for own facial femininity (r = .72), ideal facial femininity (r = .60), prediction of female facial femininity preferred by men for short- (r = .70) and long-term relationships (r = .63) and preference for male facial masculinity for short- (r = .69) and long-term (r = .72) relationships. Given the consistent and high correlation, the values for the 2 example faces for each judgment were averaged together and the average values entered as dependent variables for analysis in SPSS.”

• L336: I don’t see how perceived level of own facial masculinity is a sensible predictor in your regression, given that it was already used to compute your criterion.

Response. We acknowledge the editor’s point. Using own dimorphism to predict dissatisfaction (own - ideal dimorphism) may reflect the impact of ideal dimorphism. Therefore, we now use a different statistical approach to demonstrate the relation between the misperception of opposite-sex desires and the facial dissatisfaction index. Using ANCOVA we show that the dissatisfaction index (as the sole covariate) is related to the of misperception of opposite-sex desires. 

Additionally, in supplementary information we include ANCOVA results that show ideal dimorphism (as the sole covariate) is strongly related to the misperception of opposite-sex desires. Indeed, the larger effect size for ideal dimorphism suggests that it is mediating the relation between misperception of opposite-sex desires and the dissatisfaction index facial. Mediation analysis included in the supplementary information confirmed the mediation role of ideal dimorphism. 

Main body text. Results

“For participant women, ANCOVA (analysis of covariance) with prediction of men’s desires of female facial femininity for short- and long-term as a repeated measure and facial dissatisfaction as a covariate showed a main effect of facial dissatisfaction (F[1,69] = 6.12, p = .016, ηp2 = .081). The greater the facial femininity women predicted men desired, the higher the women’s facial dissatisfaction was. There was no main effect of relationship term (F[1,69] = 2.57, p = .113, ηp2 = .036) and no interaction between relationship term and the facial dissatisfaction index (F[1,69] = 2.29, p = .135, ηp2 = .032).

For participant men ANCOVA was conducted with prediction of women’s desires of male facial masculinity for short- and long-term relationships as a repeated measure and facial dissatisfaction as a covariate. This analysis showed a main effect of facial dissatisfaction (F[1,69] = 5.41, p < .023, ηp2 = .073). The greater the facial masculinity men predicted women desired, the higher the men’s facial dissatisfaction score was. There was a main effect of relationship term (F[1,69] = 6.23, p = .015, ηp2 = .083) reflecting men’s prediction that women preferred higher levels of facial masculinity for short- rather than long-term relationships. There was no interaction between relationship term and the facial dissatisfaction index (F[1,69] = 0.08, p = .775, ηp2 = .001).

Equivalent ANCOVAs with ideal dimorphism as a covariate show that it is strongly related to the misperception of opposite-sex desires (see Supplementary Information). Mediation analysis included in the supplementary information confirmed the mediation role of ideal dimorphism in the relation between misperception and dissatisfaction.”

Main body text. Discussion.

“Analysis in supplementary information suggested that the relationship between misperception of opposite-sex desires and facial dissatisfaction was mediated by ideal dimorphism. Misperception of desires was directly associated with exaggerated ideals and exaggerated ideals were in turn directly associated with a higher facial dissatisfaction score. The direct pathway between misperception and dissatisfaction was non-significant when both ideals and misperception were used as predictors of dissatisfaction.” 

• L499, “involved White models and predominantly White UK participants”: How does this agree with your “White ethnicity” pre-screening criterion (L191)?

Thank you for picking up this error. All participants gave current country of residence as UK. The text has been changed to read: “Our study involved White models and White participants currently resident in the UK.”

 = = = = = = = = = = = = = = = = = = = = = = = = = = = = = = 

Kind regards,

Johannes Hönekopp

Academic Editor

PLOS ONE

Journal Requirements:

https://journals.plos.org/plosone/s/file?id = wjVg/PLOSOne_formatting_sample_main_body.pdf and 

https://journals.plos.org/plosone/s/file?id = ba62/PLOSOne_formatting_sample_title_authors_affiliations.pdf

Formatting has been standardised following PLoS One recommendations

We have uploaded data to figshare. We also maintain the data deposit at OSF.

“The data that support the findings of this study are openly available on the Open Science Framework at https://osf.io/3rkxz/?view_only = 46a7dd9adae048218c328ab9a912b331 . The data are also available at https://figshare.com/articles/dataset/Misperception_of_the_desirability_of_sexual_dimorphism/26424307?file = 48071893”

3. We note that Figures 1, 3 and 5 includes an image of a participant in the study. 

As per the PLOS ONE policy (http://journals.plos.org/plosone/s/submission-guidelines#loc-human-subjects-research) on papers that include identifying, or potentially identifying, information, the individual(s) or parent(s)/guardian(s) must be informed of the terms of the PLOS open-access (CC-BY) license and provide specific permission for publication of these details under the terms of this license. Please download the Consent Form for Publication in a PLOS Journal (http://journals.plos.org/plosone/s/file?id = 8ce6/plos-consent-form-english.pdf). The signed consent form should not be submitted with the manuscript, but should be securely filed in the individual's case notes. Please amend the methods section and ethics statement of the manuscript to explicitly state that the patient/participant has provided consent for publication: “The individual in this manuscript has given written informed consent (as outlined in PLOS consent form) to publish these case details”. 

Response. We note that the illustrations of facial images are not of real individuals. In Figure 1 the images contain the shape, colour and texture o

---

## [Editor Report · Decision Letter 1]

8 Sep 2024

Misperception of the facial appearance that the opposite-sex desires

PONE-D-24-22275R1

Dear Dr. Perrett,

We’re pleased to inform you that your manuscript has been judged scientifically suitable for publication and will be formally accepted for publication once it meets all outstanding technical requirements.

Kind regards,

Johannes Hönekopp

Academic Editor

PLOS ONE
---

## [Editor Report · Acceptance letter]

16 Sep 2024

PONE-D-24-22275R1 

PLOS ONE

Dear Dr. Perrett, 

I'm pleased to inform you that your manuscript has been deemed suitable for publication in PLOS ONE. Congratulations! Your manuscript is now being handed over to our production team.

Kind regards, 

on behalf of

Dr. Johannes Hönekopp 

Academic Editor

PLOS ONE